# Adrenal Insufficiency with Anticancer Tyrosine Kinase Inhibitors Targeting Vascular Endothelial Growth Factor Receptor: Analysis of the FDA Adverse Event Reporting System

**DOI:** 10.3390/cancers14194610

**Published:** 2022-09-22

**Authors:** Emanuel Raschi, Michele Fusaroli, Valentina Giunchi, Andrea Repaci, Carla Pelusi, Veronica Mollica, Francesco Massari, Andrea Ardizzoni, Elisabetta Poluzzi, Uberto Pagotto, Guido Di Dalmazi

**Affiliations:** 1Pharmacology Unit, Department of Medical and Surgical Sciences (DIMEC), Alma Mater Studiorum–University of Bologna, 40126 Bologna, Italy; 2Division of Endocrinology and Diabetes Prevention and Care Unit, IRCCS Azienda Ospedaliero–Universitaria di Bologna, 40138 Bologna, Italy; 3Department of Medical and Surgical Sciences (DIMEC), Alma Mater Studiorum–University of Bologna, 40138 Bologna, Italy; 4Medical Oncology, IRCCS Azienda Ospedaliero–Universitaria di Bologna, 40138 Bologna, Italy; 5Department of Experimental, Diagnostic and Specialty Medicine, Policlinico S. Orsola-Malpighi, Alma Mater Studiorum—University of Bologna, 40138 Bologna, Italy

**Keywords:** VEGFR, adrenal insufficiency, pharmacovigilance, disproportionality analysis, FAERS

## Abstract

**Simple Summary:**

This real-world post-marketing research described clinical features of adrenal insufficiency with anticancer drugs targeting vascular endothelial growth factor receptor reported to the Food and Drug Administration pharmacovigilance database. A robust signal emerged for multi-targeted tyrosine kinase inhibitors, especially in combination with immunotherapy, namely checkpoint inhibitors in renal cancer. These signals should promote both prospective research and a multidisciplinary proactive monitoring by healthcare professionals. These findings strengthen the role of timely pharmacovigilance to detect and characterize post-marketing adverse events of special interest, thus supporting patient care.

**Abstract:**

*Background*: We described clinical features of adrenal insufficiency (AI) reported with tyrosine kinase inhibitors (TKIs) targeting vascular endothelial growth factor receptor (VEGFR) in the Food and Drug Administration Adverse Event Reporting System (FAERS). *Methods*: Reports of AI recorded in FAERS (January 2004–March 2022) were identified through the high-level term “adrenal cortical hypofunctions”. Demographic and clinical features were inspected, and disproportionality signals were detected through the Reporting Odds Ratio (ROR) and Information Component (IC) with relevant 95% confidence/credibility interval (CI), using different comparators and adjusting the ROR for co-reported corticosteroids and immune checkpoint inhibitors (ICIs). *Results*: Out of 147,153 reports with VEGFR-TKIs, 314 cases of AI were retained, mostly of which were serious (97.1%; hospitalization recorded in 44.9%). In a combination regimen with ICIs (43% of cases), VEGFR-TKIs were discontinued in 52.2% of the cases (26% as monotherapy). The median time to onset was 72 days (IQR = 14–201; calculated for 189 cases). A robust disproportionality signal emerged, also in comparison with other anticancer drugs (ROR = 2.71, 95%CI = 2.42–3.04; IC = 0.25, 95%CI = 0.07–0.39). Cabozantinib, sunitinib and axitinib generated robust disproportionality even after ROR adjustment. *Conclusions*: We call pharmacologists, internists, oncologists and endocrinologists to raise awareness of serious AI with VEGFR-TKIs, and to develop dedicated guidelines, especially for combination regimens with immunotherapy.

## 1. Introduction

Angiogenesis inhibitors targeting the vascular endothelial growth factor and relevant receptor (VEGF/VEGFR) represent an established anti-cancer strategy in the management of various solid tumors, either alone or in combination with chemotherapy/immunotherapy. Three main pharmacological classes can be identified, including tyrosine kinase inhibitors (TKIs), an essential therapeutic option especially in renal cell carcinoma and thyroid neoplasms, monoclonal antibodies (mAbs) such as bevacizumab in metastatic colorectal carcinoma, and VEGF soluble decoy receptor capturing free VEGF such as the fusion protein aflibercept [1]. In contrast to mAbs, VEGFR-TKIs inhibit additional tyrosine kinases, including, among others, platelet-derived growth factor receptor (PDGFR), fibroblast growth factor receptors (FGFR 1–4), platelet-derived growth factor receptor-alpha (PDGFRα), and mast/stem cell growth factor receptor (KIT), thus making them multi-targeted TKIs [2].

Endocrine toxicity with VEGFR-TKIs is an emerging area of interest: understanding the mechanisms affecting endocrine vasculatures is crucial to minimizing adverse events (AEs) associated with their use [3]. In particular, fatigue, diarrhea, and weight loss represent most frequent AEs. Notably, chronic severe fatigue in oncology may be a cancer-related complication but also possibly subtend underlying adrenal dysfunctions, thus leading to drug discontinuation. However, the effects of VEGFR inhibitors on adrenal function are incompletely characterized [4,5]. To the best of our knowledge, only a few case reports/series have described adrenal insufficiency (AI) during bevacizumab [6,7], sunitinib [8,9], pazopanib [10], lenvatinib [11], and vandetanib therapy [12,13]. Moreover, the Food and Drug Administration (FDA) recently warned in the label (January 2021) that, when cabozantinib is used in combination with nivolumab, primary or secondary AI may occur (https://www.accessdata.fda.gov/drugsatfda_docs/label/2021/208692s010lbl.pdf accessed on 21 September 2022). However, case reports can only provide a partial epidemiological perspective.

The analysis of spontaneous reporting databases, such as the FDA Adverse Event Reporting System (FAERS), allows a broader perspective by collecting unpublished reports of AEs submitted worldwide occurring in real-world unselected subjects with comorbidities, poly-pharmacotherapy and complex anticancer combination regimens, also in the long-term; this ensures rapid detection of even rare AEs and emerging clinical entities such as AI, especially for drugs authorized through conditional accelerated approval [14]. Moreover, the analysis of pharmacodynamic data may be used to explore the underlying pathways involved in drug-induced diseases with TKIs [15,16,17].

This pharmacovigilance study aims to characterize the main features of AI with VEGFR-TKIs in FAERS, including the underlying pharmacological basis through receptor affinity data.

## 2. Materials and Methods

### 2.1. Data Sources and Study Design

FAERS is one of the largest publicly available repositories of unsolicited reports, gathering more than 20 million reports worldwide, including the US, Europe and Asia. By virtue of this comprehensive catchment area and the opportunity to access raw data for customized analyses, it was largely used to identify and characterize AEs with anticancer drugs, including VEGFR-TKIs [18,19].

We performed a retrospective disproportionality analysis of FAERS data (January 2004–March 2022) which were downloaded (https://fis.fda.gov/extensions/FPD-QDE-FAERS/FPD-QDE-FAERS.html accessed on 1 July 2022) and processed to remove duplicates (i.e., reports overlapping in key pre-specified fields, including active substance(s), AEs, event date, age, gender, reporter country, weight).

The protocol was pre-registered on the Open Science Framework (https://osf.io/bmgc9/ created on 5 July 2022) and analyses were performed through the open-source R software (version 4.0.2; 22 June 2022).

### 2.2. Definition of Cases and Drugs of Interest

In the FAERS database, AEs are classified according to the Medical Dictionary for Regulatory Activities (MedDRA) terminology in terms of signs/symptoms, called preferred terms (PTs). AEs of interest were identified using PTs under the high-level term “adrenal cortical hypofunctions”, in line with a recent study [20].

We identified 10 FDA approved TKIs having VEGFR as the main target: lenvatinib, sorafenib, vandetanib, cabozantinib, sunitinib, regorafenib, axitinib, pazopanib, tivozanib, and nintedanib. We also analyzed three mAbs targeting VEGF (bevacizumab, ranibizumab, ramucirumab) and the fusion protein aflibercept. Immune checkpoint inhibitors (ICIs) served as a positive control, considering their recognized association with AI [20,21]. Cyclin-Dependent Kinase (CDK)-TKIs (abemaciclib, palbociclib, ribociclib) and Janus Kinase (JAK)-TKIs (upadacitinib, ruxolitinib, tofacitinib, baricitinib) served as negative controls.

### 2.3. Descriptive Analysis

Reports of AI were first described in terms of patient demographics (sex, age, country, type of reporter) and clinical features: concomitant drugs, comorbidities, fatality proportion (i.e., death reported as the outcome), seriousness (focusing on events resulting in hospitalization), latency (i.e., time to onset), discontinuation and positive dechallenge (clinical improvement after the offending agent is suspended).

Differences in categorical variables were assessed using a chi-squared test of independence performed on a 2 × 2 contingency table with Yates’ continuity correction. Differences in continuous variables were assessed using a Kruskal-Wallis test. The Holm Bonferroni method was used to correct the *p*-values for multiple comparisons. Significance was assumed when the *p* value, corrected for multiple comparisons using the Holm-Bonferroni method, was less than 0.05.

### 2.4. Disproportionality Analysis

We performed the so-called case/non-case approach, a validated concept in pharmacovigilance, to assess whether suspected adrenal AEs (cases) are differentially reported with VEGF/VEGFR inhibitors as compared to other AEs (non-cases) [22]. If an imbalance in the number of cases vs. non-cases emerges, a disproportionality signal can be claimed, thus informing physicians to proactive monitoring [23]. We calculated two different disproportionality measures to reduce the likelihood of false positives:(a)the frequentist reporting odds ratio (ROR), deemed statistically significant by the lower limit of the 95% of the Confidence Interval (CI) >1, with at least five cases, using the Bonferroni correction to reduce the likelihood of detecting spurious associations related to multiple testing [20].(b)the Bayesian Information Component (IC), deemed significant by the 95% credibility interval, IC0_25_ > 0, which is more accurate in case of low number of cases [24].

Both approaches were first performed using all other drugs in the FAERS database as a comparator, a common exploratory disproportionality analysis. To assess the robustness of disproportionality signals and account for potential confounders of the drug-event association, we planned *a priori* a sensitivity disproportionality analysis using anticancer drugs as a comparator (to reduce confounding by indication and provide a clinical perspective), and adjusting the ROR for possible drug-related confounders, namely concomitant corticosteroids and immunotherapy [20].

### 2.5. Exploring the Underlying Pharmacological Basis

We retrieved affinity data from the Chembl Database (https://www.ebi.ac.uk/chembl/ accessed on 5 July 2022), a consolidated source of pharmacodynamic data [25] (Appendix A). We interpolated the strength of disproportionality (IC values vs. anticancer drugs) with Homo Sapiens pKd and pIC_50_ values (= negative decadic logarithm of Kd and IC_50_ [M]). When multiple values were available, the geometrical mean was calculated.

### 2.6. Global Assessment of the Evidence

Causality assessment was carried out on the entire body of evidence by using adapted Bradford Hill Criteria used in epidemiology, which have been applied to pharmacovigilance data [25,26,27]: biological plausibility, strength, consistency, specificity, coherence, and analogy. The biological gradient was not analyzed due to missing data.

## 3. Results

### 3.1. Descriptive Analysis

Overall, 12,030,756 reports were retained after FAERS processing, of which 147,153 were for VEGFR-TKIs (1.22%). A total of 9649 reports of adrenal hypofunctions were found (mainly AI, N = 6402); in 314 cases, at least one VEGFR-TKI was reported (3.25% of total adrenal hypofunction reports in FAERS, and 0.21% within VEGFR-TKIs), especially cabozantinib (74), lenvatinib (67), sunitinib (62), axitinib (58), lenvatinib (39), pazopanib (26), sorafenib (22), and nintedanib (10). We also found 130 AI cases with mAbs, almost exclusively by bevacizumab.

We observed reporting differences between cases and non-cases (Table 1 and Table 2). Concerning concomitant drugs, a higher proportion of cases described the concomitant use of ICIs (43 vs. 6%), glucocorticoids (32 vs. 8%), proton pump inhibitors (23 vs. 11%) and thyroid hormones (20 vs. 8%). Concerning co-reported events, a higher proportion of cases co-reported asthenic conditions (22 vs. 19%), non-infective diarrhea (20 vs. 17%), appetite disorders (12 vs. 9%), vascular hypotensive disorders (eight vs. 1%), vascular hypertensive disorders (12 vs. 6%) and thyroid hypofunction disorders (10 vs. 1%). Renal neoplasm represented the most frequently reported indication of AI cases (58 vs. 39%).

Of note, cases with VEGFR-TKIs were mainly reported by clinicians (57.8 vs. 36.6% for non-cases), and classified as serious (97.1%, with hospitalization recorded in 44.9%). A peak was observed in 2021 (89 reports in 2021 vs. 42 reports in 2020). Non-cases, while presenting a reporting peak in 2021, had an almost constant growth in the period under study. VEGFR-TKIs were almost exclusively reported as suspect (95.9%). Discontinuation and dechallenge were recorded in 37.6% and 34.1% of the cases, respectively; of note, relevant rates were two-fold higher in the combination regimen with ICIs as compared to monotherapy (52.2% vs. 26.7% and 47.8% vs. 23.9%, respectively). The median time to onset was 72 days [interquartile range (IQR) = 14–201; calculated for 189 cases], higher in combination with ICIs (122 days; IQR = 31–236).

### 3.2. Disproportionality Analysis

Increased reporting of AI emerged for VEGF/VEGFR inhibitors as a class when compared to all other drugs recorded in the FAERS database, with consistent disproportionality for both frequentist (ROR = 2.22, 95%CI = 2.02–2.44) and Bayesian (IC = 1.11, 95%CI = 0.95–1.22) approaches. TKIs and mAbs generated significant disproportionality as a class, especially cabozantinib, lenvatinib, sunitinib, axitinib and bevacizumab. ICIs were confirmed as a positive control (ROR = 26.36, 95%CI = 24.75–28.05; IC = 4.5, 95%CI = 4.4–4.57), whereas JAK inhibitors and CDK4/6 inhibitors were confirmed as negative controls (Figure 1, Figure 2 and Appendix A).

When the analysis was performed within anticancer agents, disproportionality remained statistically significant for TKIs (ROR = 1.21, 95%CI = 1.07–1.36; IC = 0.25, 95%CI = 0.07–0.39), specifically cabozantinib, lenvatinib, and axitinib. In the sensitivity analysis (ROR adjusted for concomitant ICIs and corticosteroids), disproportionality remained statistically significant for axitinib (ROR = 1.61, 95%CI = 1.23–2.08), cabozantinib (1.41, 1.10–1.76) and sunitinib (2.16, 1.66–2.76) (Appendix A).

### 3.3. Global Assessment and Pharmacological Basis

Globally, Bradford Hill criteria were fulfilled, as indicated by the strength of disproportionality and its consistency throughout the analyses, temporal relationship, and biological plausibility, thus supporting a likely causal association for TKIs (Table 3).

The interpolation of ROR and affinity towards different receptors highlighted a common antagonism for VEGFR-1/2/3, and PDGFR-alpha/beta, especially shared by TKIs with significant disproportionality (Figure 3).

## 4. Discussion

Targeting VEGF/VEGFR has emerged as an effective therapeutic strategy in different cancers, especially in combination with ICIs. In fact, TKIs exert immunomodulatory properties that may shift the tumor microenvironment from immunosuppressive to immune-permissive [28]. However, different life-threatening off-target toxicities have been described, including thyroid dysfunctions, cardiovascular events, bleeding, proteinuria/renal impairment, hepatotoxicity, gastrointestinal perforations, metabolic and wound healing complications with an impact on quality of life and monitoring strategies [29].

This study is the largest collection of AI cases attributed to VEGF/VEGFR inhibitors from a worldwide pharmacovigilance database so far, and raised awareness on the reporting of serious events. Apart from recently approved drugs such as tivozanib, there is a considerable amount of data for other agents with a consolidated time on the market.

Our study raised debate on whether AI should be considered a class effect of VEGFR-TKIs. Notably, we observed reporting differences between mAbs and TKIs: the former did not consistently generate disproportionality signals, and only bevacizumab was reported in a significant number of cases; the latter were associated with a consistent and robust increased reporting, even when accounting for potential confounders, thus suggested the existence of a plausible cause-effect relationship for VEGFR-TKIs, as corroborated by the global evaluation through Bradford Hill criteria.

Taken together, these data call for constant epidemiological surveillance and urgent clarification by means of large population-based studies. Although actual incidence and prevalence cannot be derived from spontaneous reporting, the fact that VEGFR-TKIs were reported in 314 cases over the past few years (3.25% of total adrenal hypofunction reports in FAERS) suggests that the estimated AI risk is not so rare, and we advocate for an update of the relevant summaries of product characteristics where AI is not mentioned. Of note, only the FDA warned in the product information that cabozantinib might increase the risk of AI when combined with nivolumab.

Different demographic and clinical features emerged from this study, including large male preponderance, seriousness (in nearly half of the cases, was hospitalization recorded), severity (in more than 1/3 of the cases discontinuation was needed), and frailty (higher number of concomitant drugs as compared to non-AI reports). The selective reporting of serious events suggested that our data only represented the “tip of the iceberg”, since non-serious cases are unlikely to be diagnosed and submitted to pharmacovigilance databases, thus calling for dedicated prospective clinical pathways to capture subclinical AI and clarify the actual epidemiological burden. Moreover, the exponential increase in the number of AI reports (as compared to other events), especially in 2021, is noteworthy and may be ascribable to various reasons, including the increasing uptake in clinical practice and expanding therapeutic indications, especially in a combination regimen with immunotherapy.

The available clinical evidence on AI by VEGFR-TKIs is scant, jeopardized and limited to cases reports and small case series [4,5], including a recent pharmacovigilance study on axitinib [30]. In a small prospective study, the analysis of basal and stimulated adrenal function in 12 patients receiving lenvatinib and vandetanib for advanced radioiodine refractory differentiated or medullary thyroid cancer, respectively, revealed a gradual ACTH increase with normal cortisol levels in 10 patients complaining of fatigue. Notably, patients diagnosed with primary AI were treated with cortisone acetate replacement therapy, and fatigue improved [13]. As expected, we found a large proportion of AI reports with co-reported asthenic conditions, diarrhea and vascular events, which should raise suspicion by clinicians to early diagnose an underlying adrenal hypofunction.

The issue of combination regimen (VEGFR-TKIs plus ICIs) deserves further discussion, especially in metastatic renal cell carcinoma, where this combination is the preferred first-line approach in patients with favorable, intermediate and poor risk disease [31]. VEGFR-TKIs and ICIs share different overlapping toxicities, possibly including AI. However, pivotal trials were not designed to test the actual contribution of individual TKIs, thus making it challenging to unravel the inherent risk of AI. For instance, cabozantinib plus nivolumab was compared with sunitinib in a CheckMate 9ER trial [32]; grade ≥ 3 AI occurred in 1.9% of subjects (vs. zero event), with a median time to onset of 37.3 weeks (highly variable; IQR = 4.1–76.7; n = 320) [33]. Minor imbalances were noted in a KEYNOTE-426 trial for pembrolizumab plus axitinib (0.7% vs. 0.2% for sunitinib) [34], and a CLEAR trial for lenvatinib plus pembrolizumab (1.1% vs. 0% for sunitinib) [35]. Our real-world data found a median latency of three months for the combination regimen, with remarkable high rates of discontinuation and dechallenge, thus suggesting the importance of implementing dedicated guidelines and screening protocols (ACTH test) through a close collaboration among pharmacologists, internists, oncologists and endocrinologists [4,36,37].

A recent pharmacovigilance study on FAERS (up to September 2020) specifically addressed whether the various combination regimens of VEGFR-TKI with ICIs increased the reporting of AI, as compared to monotherapies, using various approaches and models. No increased reporting was observed across combinations and analyses, except for bevacizumab when combined with atezolizumab, thus failing to demonstrate the existence of a pharmacodynamic interaction, namely additive or synergistic effect [36]. Disproportionality signals were found for cabozantinib, axitinib and lenvatinib, in line with our results, thus confirming the hypothesis that VEGFR-TKIs *per se* likely causes AI. Unraveling the contribution of VEGFR-TKIs vs. ICIs represents the next clinical and research challenge: actual understanding of pathophysiology would allow proper management (i.e., the interruption of a single drug or the combination).

From a mechanistic viewpoint, one important insight can be derived from the appraisal of pharmacodynamics (receptor affinities) with pharmacovigilance data: we put forward the inner role of VEGFR-1/2/3, highly expressed in the fenestrated capillaries of endocrine organs. Notably, axitinib, cabozantinib and lenvatinib emerged with the strongest association and also possess the highest potency towards VEGFR [38]. We encourage regular surveillance by clinicians and pharmacovigilance experts on these drugs, including the third-generation agent tivozanib (higher selectivity towards VEGFR and favorable pharmacokinetics), recently approved for advanced renal cell carcinoma and under additional regulatory monitoring.

We acknowledge inherent limitations of pharmacovigilance data, which do not allow to infer causality (ROR and IC are not risk measures); reports may be incomplete, with missing information (e.g., laboratory and other clinical elements to validate the diagnosis). Moreover, incidence cannot be estimated due to under-reporting and lack of population exposure. The lack of disproportionality should not be interpreted as a safety endorsement, since disproportionality measures are interdependent (a commonly reported AE may mask rare AEs) and the choice of comparator may have a substantial impact. Moreover, the various drugs have different marketing approvals and evidence-based treatment recommendations have changed over recent years. All of these aspects may explain why pazopanib and sorafenib were not associated with significant disproportionality notwithstanding the non-negligible number of reported AI cases. Finally, the contribution of additional drugs with underlying adrenal liability cannot be excluded, including antimicrobials, hormone therapy and drugs for Cushing or adrenocortical cancer diagnosis/treatment.

Nonetheless, several strengths can be identified. We used a large-scale publicly accessible pharmacovigilance database, supporting the generalizability of the results, and contributed to the cumulative knowledge about the safety of VEGFR-TKIs in an unselected real-world population, an evolving clinical issue. There is no reason *a priori* to support the existence of confounding by indication or notoriety bias (i.e., increased reporting following media attention or regulatory measures), although channeling bias (i.e., selective prescription towards more severe patients) cannot be ruled out. Moreover, we employed several sensitivity analyses to assess the robustness of results, accounting for potential co-reporting biases and using anticancer drugs as comparator, thus providing a clinical perspective. The fact that the signal for VEGFR-TKIs emerged even without accounting for the potential masking effect by previously known and largely reported AEs further strengthened the impact of AI in the overall risk-benefit evaluation. Our global pharmacovigilance assessment, encompassing disproportionality, approaches and Bradford Hill criteria evaluation consistently support the notion of a plausible association.

## 5. Conclusions

Although causality cannot be firmly inferred, this study found a plausible association between VEGFR-TKIs and AI; the selective reporting of serious events should raise awareness by clinicians and suggested the existence of subclinical AI cases. We call for a tight multidisciplinary collaboration between pharmacologists, internists, oncologists and endocrinologists to: (a) proactively monitor the occurrence of AI with VEGFR-TKIs; and (b) implement dedicated guidelines, especially for a combination regimen with immunotherapy.

Considering the evolving use and recent marketing approvals of VEGFR-TKIs, pharmacovigilance plays a key role in promoting targeted clinical surveillance and safer prescribing, especially for rare AEs of special interest such as AI.

## Figures and Tables

**Figure 1 cancers-14-04610-f001:**
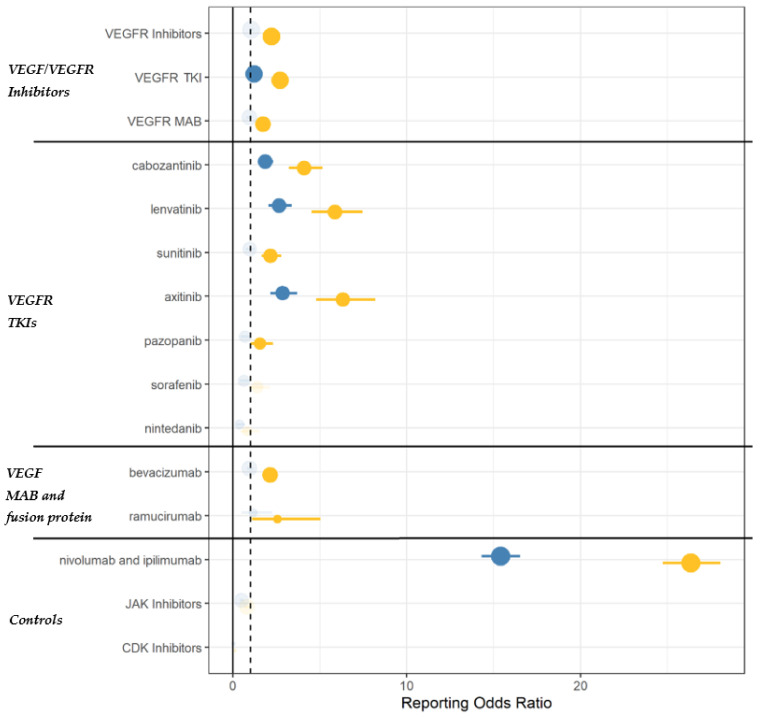
Disproportionality analysis through the Reporting Odds Ratios (RORs). Yellow and blue dots represent ROR estimates that were calculated using all drugs in the database, and anticancer drugs as a comparator, respectively. Estimates are proportional to the number of cases. VEGFR: vascular endothelial growth factor receptor; TKI: tyrosine kinase inhibitors; MAB: monoclonal antibodies.

**Figure 2 cancers-14-04610-f002:**
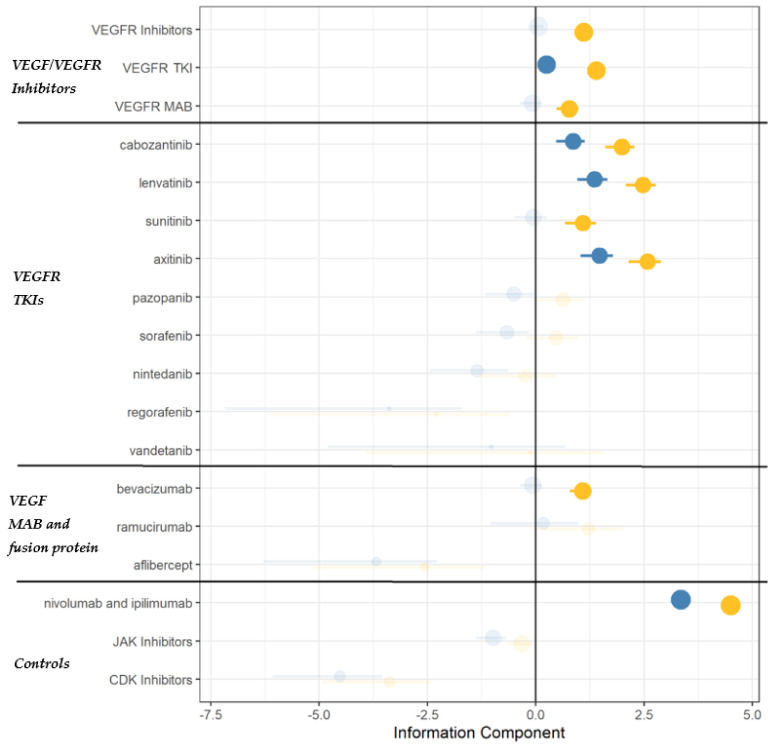
Disproportionality analysis through the Information Component (IC). Yellow and blue dots represent IC estimates that were calculated using all drugs in the database, and anticancer drugs as a comparator, respectively. Estimates are proportional to the number of cases. VEGFR: vascular endothelial growth factor receptor; TKI: tyrosine kinase inhibitors; MAB: monoclonal antibodies.

**Figure 3 cancers-14-04610-f003:**
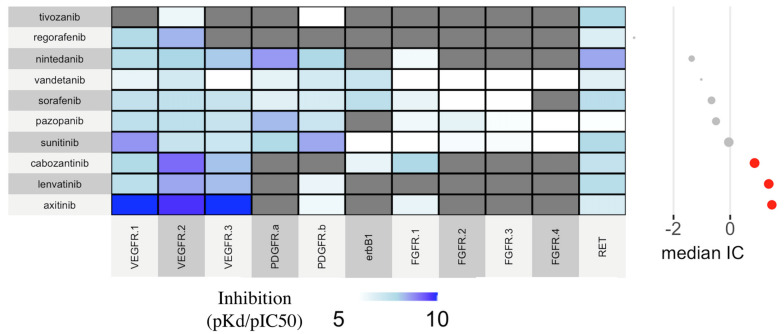
Heatmap showing the extent of VEGFR-TKIs affinities in relation with disproportionality analysis. The intensity of the cold tone (antagonism) is proportional to the receptor affinity (pKd/pIC_50_). Grey color code indicates missing pharmacodynamic data. The adjacent plot shows significant (in red) and non-significant (in grey) disproportionalities using anticancer drugs as a comparator (information component—IC).

**Table 1 cancers-14-04610-t001:** Demographic data on adrenal insufficiency (AI) with VEGFR-TKIs.

	AI (Cases) (314)		Other Reports (Non-Cases) (146,839)		*p*-Value
Demographic Features	N	%	N	%	
Sex					0.687
Female	107	37.4	48,813	37.8	
Male	179	62.6	87,591	64.2	
Missing	28	(-) ^1^	10,435	(-) ^1^	
Reporter Country					0.003
North America	122	38.9	87,023	59.3	
Europe	72	22.9	24,194	16.5	
Asia	106	33.8	27,499	18.8	
South America	10	3.2	6151	4.2	
Oceania	4	1.3	1282	0.9	
Africa	0	-	535	0.4	
Missing	0	(-) ^1^	155	(-) ^1^	
Reporter Qualification					0.003
Consumer	42	13.9	58,546	41.3	
Healthcare professionals	33	10.9	7098	5.0	
Lawyer	0	-	12	<0.1	
Medical doctor	175	57.8	47,730	33.6	
Other	41	13.5	18,172	12.8	
Pharmacists	12	4.0	10,368	7.3	
Missing	11	(-) ^1^	4913	(-) ^1^	
Age					0.687
Median years [IQR] ^2^	66 [60–73]		66 [58–73]		
Missing	71		40,354		
Reporting Year					0.003
≤2015	35	13.3	31,721	26.8	
2016	11	4.2	10,520	8.9	
2017	18	6.8	12,654	10.7	
2018	31	11.8	15,394	13.0	
2019	43	16.4	15,694	13.3	
2020	54	20.5	15,209	12.9	
2021	71	27.0	16,951	14.3	
Missing	51		28,666		

^1^ Percentages were calculated considering only specified values (i.e., valid percentages). ^2^ IQR: interquartile range.

**Table 2 cancers-14-04610-t002:** Clinical data on adrenal insufficiency (AI) with VEGFR-TKIs.

	AI (Cases) (314)		Other Reports (Non-Cases) (146,839)		*p*-Value
Clinical Features					
Outcome					0.003
Death	41	13.1	31,973	21.8	
Life-threatening	27	8.6	3376	2.3	
Disability	6	1.9	1412	1.0	
Required intervention	1	0.3	87	0.1	
Hospitalization	141	44.9	39,050	26.6	
Congenital anomaly	0	0	9	<0.1	
Other Serious	89	28.3	30,572	20.8	
Non serious	9	2.9	40,360	27.5	
Time to onset					0.007
Median days [IQR] ^1^	72 [14–201]		37 [9–139]		
Missing	125		74,543		
Co-reported events					<0.001
Median N. [IQR] ^1^	4 [2–8]		2 [1–5]		
Concomitant drugs					<0.001
Median N. [IQR] ^1^	5 [2–12]		2 [1–6]		
Concomitant Immunotherapy	134	42.7	8795	6.0	<0.001

^1^ IQR: interquartile range.

**Table 3 cancers-14-04610-t003:** Global assessment through adapted Bradford Hill Criteria.

Criteria	Description	Source/Method
Strength of the association	Although ROR and IC are not measures of risk, the strength of the disproportionality both in primary (vs. all other drugs) and secondary (vs. other anticancer drugs) analyses suggests a robust signal (the impact of unmeasurable confounders is likely to be negligible)	Disproportionality
Analogy	The association was also demonstrated for other anticancer drugs used as a positive control (immune checkpoint inhibitors)	Disproportionality and Literature
Biological plausibility/empirical evidence	The interpolation between disproportionality results and pharmacodynamic data (affinity for VEGFR) supports the mechanistic basis	Disproportionality and Pharmacodynamics
Consistency	Results of disproportionality approaches were consistent in sensitivity analyses	Disproportionality
Coherence	Case reports/series have been published recently, suggesting a potential association. A recent pharmacovigilance study on axitinib found a disproportionality signal for AI	Literature
Exclusion of biases/confounders ^1^	The association (statistically significant disproportionality) persisted across sensitivity analysis accounting for confounding by indication and potential co-reported bias	Disproportionality
Specificity	Pharmacovigilance data suggest TKIs to carry a stronger association as compared to mAbs. Moreover, a drug-specific effect (rather than a class-effect) cannot be excluded	Disproportionality
Temporal relationship	Notwithstanding missing data, the time to onset (delay from the first administration of the drug to the date the event was recorded) is coherent with biological and clinical notions.	Descriptive
Reversibility	Data on discontinuation and dechallenge support the reversibility	Descriptive

^1^ These items were not included in the original Bradford Hill Criteria.

## Data Availability

The datasets analyzed during the current study are available in the following resource available in the public domain: https://fis.fda.gov/extensions/FPD-QDE-FAERS/FPD-QDE-FAERS.html (accessed on 26 July 2022).

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
