# Peer review of "Adrenal Insufficiency with Anticancer Tyrosine Kinase Inhibitors Targeting Vascular Endothelial Growth Factor Receptor: Analysis of the FDA Adverse Event Reporting System"

_cancers, 2022, doi:10.3390/cancers14194610_

Round 1
Reviewer 1 Report
Well written and and thorough analysis regarding adrenal insufficiency with tyrosine kinase inhibitor targeting VEGF receptor. Please check for minor language error. Overall paper is up to the level for publication.
Reviewer 2 Report
The current study focused on explaining clinical features of adrenal insufficiency (AI) reported with the usage of tyrosine kinase inhibitors (TKIs) targeting vascular endothelial growth factor receptor (VEGFR) in the 30 Food and Drug Administration Adverse Event Reporting System (FAERS) is a well-designed pharmacovigilance study.
The study explains the importance of reporting all the possible adverse effect could be caused due to tyrosine kinase inhibitor and effectively creates the awareness in the medical field in exploring and promptly reporting the all the possible toxicological effect of the drugs which could be ignored but could be fatal.
I recommend this article for publication.
Thank You
Reviewer 3 Report
Several aspects of this work need clarification:
1. Taking into consideration the low incidence of AI with VEGFR-TKIs, the authors should explain in a more convincing way why they studied this.
2. If I understand correctly, the authors included cases with VEGFR-TKI+ICI combinations in their study cohort. This is counterintuitive. They should repeat the analysis with VEGFR-TKI pure population
Round 2
Reviewer 3 Report
Authors have addressed my comments satisfactorily.